# Language Guided Visual Question Answering: Elevate Your Multimodal Language Model Using Knowledge-Enriched Prompts

**Deepanway Ghosal**[†]**, Navonil Majumder**[†]**, Roy Ka-Wei Lee**[†]**,**
**Rada Mihalcea**[△]**, Soujanya Poria**[†]
[†] Singapore University of Technology and Design, Singapore
[△] University of Michigan, USA
{deepanway_ghosal}@mymail.sutd.edu.sg
{navonil_majumder, roy_lee, sporia}@sutd.edu.sg
mihalcea@umich.edu

## Abstract

Visual question answering (VQA) is the task of answering questions about an image. The task assumes an understanding of both the image and the question to provide a natural language answer. VQA has gained popularity in recent years due to its potential applications in a wide range of fields, including robotics, education, and healthcare. In this paper, we focus on knowledge-augmented VQA, where answering the question requires commonsense knowledge, world knowledge, and reasoning about ideas and concepts not present in the image. We propose a multimodal framework that uses language guidance (LG) in the form of rationales, image captions, scene graphs, etc to answer questions more accurately. We benchmark our method on the multi-choice question-answering task of the A-OKVQA, Science-QA, VSR, and IconQA datasets using CLIP and BLIP models. We show that the use of language guidance is a simple but powerful and effective strategy for visual question answering. Our language guidance improves the performance of CLIP by 7.6% and BLIP-2 by 4.8% in the challenging A-OKVQA dataset. We also observe consistent improvement in performance on the Science-QA, VSR, and IconQA datasets when using the proposed language guidances. The implementation of LG-VQA is publicly available at https://github.com/declare-lab/LG-VQA.

## 1 Introduction

Visual understanding is one of the most complex tasks in artificial intelligence. Among the many challenges associated with it, image question answering has been formulated as a task that tests the ability of a system to understand the elements of an image in a way similar to how humans interact with images. This task involves creating models that can accurately answer questions based on the content of an image. While significant progress has been made in image question answering (Wang et al., 2022a; Chen et al., 2022), most

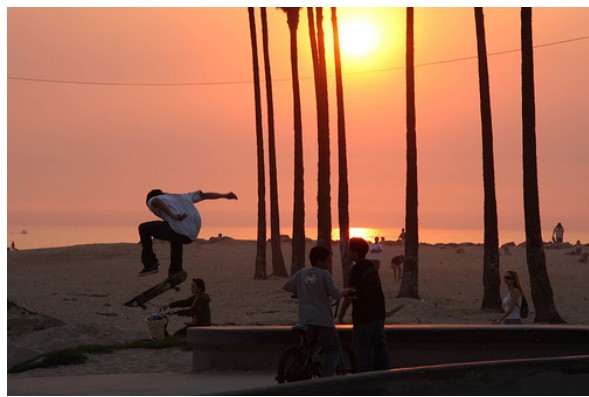

Figure 1: The above image and the question *At what time of day are the skateboarders probably skating on the beach?* are given. In our LG-VQA framework, we generate the caption: *a group of skate boarders in front of palm trees at sunset* and the question constrained rationale: *the sun is just beginning to set behind the sand* to provide the answer: *sunset*.

of the existing approaches focus solely on analyzing the visual features associated with the image or the linguistic content or representation of the question (and possibly candidate answers), without utilizing additional sources of guidance. However, incorporating external guidance into these models has the potential to improve their performance and enhance their understanding of visual content.

In this paper, we propose a multimodal framework, LG-VQA that leverages language guidance to improve the accuracy of image question-answering systems. Language guidance is sourced from elements such as image captions, scene graphs, and rationales created in response to questions. These sources serve to enrich textual instructions with valuable knowledge. Our method is evaluated on the multi-choice A-OKVQA dataset (Schwenk et al., 2022). We choose A-OKVQA specifically because the questions in this dataset require a broad base of commonsense and world knowledge to answer. We also evaluate our method on the challenging ScienceQA (Lu et al.,

2022), Visual Semantic Reasoning (VSR) (Liu et al., 2023), and IconQA (Lu et al., 2021) datasets. With the advent of large-scale multimodal pre-training (Wang et al., 2022a,b), the performance in the commonly used VQA and VQA v2 datasets (Antol et al., 2015; Goyal et al., 2017) has saturated.

The datasets considered in this paper provide a more challenging test bed for VQA where factual, commonsense, physical, scientific, and visual knowledge are required, going beyond the usual image recognition, attribute detection tasks that are generally found in the VQA, VQA v2 datasets. Previously Shah et al. (2019); Wu et al. (2017); Zheng et al. (2021) have used concrete knowledge bases such DBPedia, Freebase, or named entity knowledge graphs in VQA. We instead show that simple language guidance without using knowledge graphs is also a powerful technique for improving VQA performance in various challenging datasets.

We benchmark our approach with the CLIP (Radford et al., 2021) and BLIP-2 (Li et al., 2023) models and show significant improvements over those, demonstrating that incorporating language guidance can be an effective strategy for visual question answering.

## 2 Background

We use the following two strong multi-modal models in our framework:

**CLIP** (Radford et al., 2021) is a model for learning transferable visual representations from natural language supervision. CLIP is trained on 400 million (image, text) pairs to learn a joint embedding space where image representations and their corresponding text representations are close to each other. We use the CLIP model with the ViT-L/14 image encoder (Dosovitskiy et al., 2020) and the GPT-2 text encoder (Radford et al., 2019). CLIP encodes the image and the text separately through the corresponding encoders. Then, a normalized dot product between the encoded vectors provides the image-text matching score. We denote the image and text encoder as $I_{CLIP}$ and $T_{CLIP}$. Given an image $img$ and text instance $txt$, the matching score is computed as follows:

$$
\begin{aligned}
I &= \text{l2\_normalize}(I_{CLIP}(img)) \\
T &= \text{l2\_normalize}(T_{CLIP}(txt)) \\
score &= e^t * (I \cdot T) = e^t * \sum_{i=1}^{m} I_m T_m
\end{aligned}
\tag{1}
$$

where $t$ is a learned temperature parameter for scaling the dot product similarity. We use this formulation of the (image, text) matching methodology for CLIP models in our VQA framework.

**BLIP-2** (Li et al., 2023) bootstraps joint pre-training over image and language data with off-the-shelf frozen image encoders and frozen large language models (LLMs). A small transformer network called Query Transformer or Q-Former is trained to model the interaction between the frozen image and language models.

In the first stage of BLIP-2 pre-training, the Q-Former learns to extract visual representations from the image that is most relevant to the text using image-text contrastive learning, and image-grounded text generation tasks. In the second stage, the trained Q-Former is used to extract a set of query embeddings about the image. The query embeddings are converted into the input embedding space of the LLMs through a linear projection. The projected vector functions as the soft visual prompts that condition the LLM to generate the desired text. BLIP-2 uses paired (image, text) data for both stages of pre-training.

We denote the image encoder and Q-Former of BLIP2 as $I_{BLIP}$ and $Q_{BLIP}$. The image-text matching score is computed as follows:

$$
\begin{aligned}
I &= I_{BLIP}(img) \\
features &= Q_{BLIP}(I, txt, q) \\
score &= Proj(features)
\end{aligned}
\tag{2}
$$

where $q$ is the leaned embeddings during pre-training and $Proj$ is the projection head. Additionally, we use the FlanT5 (Chung et al., 2022) model as the LLM for image-constrained text generation. Here, the Q-Former does not take any text as input, but the LLM encoder may use a prefix text $prefix$ on which the $output$ text is conditioned:

$$
\begin{aligned}
v_{prompt} &= Q_{BLIP}(I, q) \\
encoded &= \text{FlanT5 Encoder}(v_{prompt}, prefix) \\
output &= \text{FlanT5 Decoder}(encoded)
\end{aligned}
\tag{3}
$$

We use the image-text matching methodology

and the (visual prompt, prefix) conditioned text generation in various stages of our framework.

# 3  LG-VQA Framework

## 3.1  Zero-Shot Visual Question Answering

The CLIP and BLIP-2 models show impressive zero-shot performance in various computer vision tasks. For ImageNet type classification tasks, the method generally works by creating sentence instances from the labels: *A photo of a dog*, *A photo of a cat*, *A photo of a bird*, etc. Then, similarity scores are computed between the image and all the sentence instances created from the labels. The label (e.g. *bird*) is then predicted from the sentence (e.g. *A photo of a bird*) which provided the maximum similarity score.

We apply a similar method for the zero-shot multi-choice visual question-answering task. Given image $img$, question $question$, multiple answer choices $a_1, a_2, \ldots, a_n$, we create sentences by concatenating the question and the answer $\{question, a_i\}$. The concatenated sentence providing the highest image-text matching score corresponds to the predicted answer:

$$
\begin{aligned}
txt_i &= \{question, a_i\} \quad \forall i = 1, \ldots, n \\
s_i &= match(img, txt_i) \\
best &= argmax([s_1, \ldots, s_n])
\end{aligned}
\tag{4}
$$
$$answer = a_{best}$$

## 3.2  Unguided Visual Question Answering

We use the term *unguided* to denote the conventional visual question-answering task with full fine-tuning without using any additional guidance. Here, we train a model with the image $img$, question $question$, and multiple answer choices $a_1, \ldots, a_n$ to predict the correct answer choice $a_k$.

We obtain the image-text matching score $s_i$ for each (image, question, answer) triplet from CLIP or BLIP-2. We then normalize the scores with a softmax layer across the $n$ choices:

$$
\begin{aligned}
txt_i &= \{q, a_i\} \quad \forall i = 1, \ldots, n \\
s_i &= match(img, txt_i) \\
\bar{s}_i &= \frac{e^{s_i}}{\sum_{j=1}^{n} e^{s_j}}
\end{aligned}
\tag{5}
$$

Finally, we use the cross-entropy loss to train the underlying CLIP or BLIP-2 model. Assuming that $a_k$ is the correct answer, the loss is as follows:

$$
\mathcal{L} = -\sum_{i=1}^{n} p_i log(\bar{s}_i) = -log(\bar{s}_k) \tag{6}
$$

where we denote $p_i$ to be the class label of the answer choices. The value of $p_k$ is 1 as $a_k$ is the correct answer, whereas the values of the other $p_i$ are zero as they are the incorrect answers. The loss thus simplifies to the $-log(\bar{s}_k)$ as shown in Eq. (6). The loss is equivalent to the cross-entropy loss used in multi-class classification problems.

## 3.3  Constained Inference Generation

We use the BLIP-2 model with FLAN-T5 to generate inferences (such as rationales, explanations and captions) constrained on the image $img$ and question $question$ that could potentially be helpful for the VQA task. We follow the setting specified earlier in Eq. (3). We do not use any $prefix$, and thus the FLAN-T5 model is constrained entirely upon the *visual prompt* extracted from the question and the image with the Q-Former.

We use the rationales in the A-OKVQA dataset and the explanations for the VQA and VQA v2 provided in (Li et al., 2018; Park et al., 2018) to fine-tune the BLIP-2 with FLAN-T5 model. We also use COCO Captions (Chen et al., 2015) for image captioning. We keep the visual encoder and the FLAN-T5 frozen and only tune the Q-Former network for the tasks of rationales or explanation or caption generation. We use the generated rationales, explanations, and captions as language guidance in our main model (§3.5).

## 3.4  Scene Graphs and Object Detection

We use the Relation Transformer (RelTR) network network (Cong et al., 2023) for scene graph generation from the images. RelTR uses a transformer model based on DETR (Carion et al., 2020) pretrained on the Visual Genome dataset (Krishna et al., 2017) to predict the scene graph triplets about an image. We use the RelTR model to predict scene graphs for the images. We concatenate the triplets with [SEP] token to create the scene graph guidance. We also use the UniDet (Zhou et al., 2022) and DETR models for object detection in the images. After detection, we aggregate the objects using their counts: *two dogs, one girl, three toys*. The objects detected are potentially helpful for questions that require counting or numerical reasoning.

## 3.5 Language Guided Visual Question Answering

We use the notation $guide$ to denote the various guidance strategies or their combinations. Concretely it could be either the rationales, explanations, captions, scene graphs, objects, or any of their combinations. The combination is achieved by simply concatenating the various guidances. We use the notation $txt\_guide_i$ to denote the concatenation of $\{question, a_i, guide\}$.

We follow two strategies for using the $guide$ in CLIP or BLIP-2. For CLIP, the strategy for computing the matching score is similar to Eq. (5) with $text\_guide_i$:

$$s_i = match(img, txt\_guide_i) \qquad (7)$$

Then, we use the softmax normalization with cross-entropy loss specified in Eq. (6) for training. CLIP was originally trained on a maximum length of 77 input tokens. We extend the maximum length to 512 tokens to incorporate $guide$ as part of the input and update these positional embeddings during training.

For BLIP-2, we found that passing $txt_i$ and $txt\_guide_i$ through two forward passes followed by feature merging to be the optimal strategy:

$$\begin{aligned}
x_1 &= Q_{BLIP}(I, txt_i, q) \\
x_2 &= Q_{BLIP}(I, txt\_guide_i, q) \\
features &= |x_1, x_2, x_1 - x_2, x_1 * x_2| \\
s_i &= Proj(features)
\end{aligned} \qquad (8)$$

The model is trained with the cross-entropy loss with softmax normalized scores over the choices. We train the Q-Former of BLIP-2, the projection layer $Proj$ and keep the visual encoder frozen. We found that fine-tuning the visual encoder is computationally expensive, with marginal improvement in performance.

## 4 Experiments

**Dataset Used** We use the multi-choice question answering setup of the A-OKVQA (Schwenk et al., 2022) dataset for evaluating our VQA framework. Each instance has four possible answer choices, among which only one is correct. We report the scores in the validation dataset as the test set labels are not available. We also report results for the easy and hard subset as specified in the dataset. The dataset has a total of 25K instances.

| Mode | CLIP | | | BLIP-2 | | |
|---|---|---|---|---|---|---|
| | Overall | Easy | Hard | Overall | Easy | Hard |
| Zero-Shot | 58.52 | 58.98 | 51.43 | 64.98 | 65.95 | 50.00 |
| No Guidance | 68.30 | 68.74 | 61.43 | 75.02 | 76.09 | 58.57 |
| LG-VQA | | | | | | |
|   Rationale | 74.11 | 74.72 | 64.71 | 76.77 | 77.77 | 61.43 |
|   Explanation | 68.21 | 68.93 | 57.14 | 76.16 | 77.02 | 62.86 |
|   Captions | 69.08 | 69.67 | 60.00 | 76.68 | 77.77 | 60.00 |
|   Scene Graph | 68.03 | 66.23 | **65.71** | 76.61 | 77.51 | 62.86 |
|   Objects | 67.77 | 68.56 | 55.71 | 75.94 | 77.07 | 58.57 |
|   All | **75.98** | **76.65** | **65.71** | **79.83** | **80.47** | **70.00** |

Table 1: Results on the A-OKVQA validation set. We report VQA accuracy scores for zero-shot, unguided, and various kinds of guidance for the overall, easy, and hard sets. Scores are average of three runs.

We also experiment with the ScienceQA (Lu et al., 2022), Visual Semantic Reasoning (VSR) (Liu et al., 2023), and IconQA (Lu et al., 2021) datasets. ScienceQA is collected from elementary and high school science curricula containing 10k questions with image context. The number of answer choices varies between 2 and 5, among which only one is correct. Visual Spatial Reasoning (VSR) is a corpus of caption-image pairs with true/false labels. Each caption describes the spatial relation of two individual objects in the image. The task is to determine if the caption correctly describes the image. We use the true/false labels as the two answer choices for multi-choice question answering. IconQA is a dataset for diagram understanding and cognitive reasoning in real-world diagram word problems. We use the multi-text-choice subtask which has a total of 31k instances.

**Results for A-OKVQA** We report the results for the overall set and the easy, hard subsets for A-OKVQA in Table 1. In the first two rows of the Table 1, we report the results in the baseline settings: i) zero-shot performance using the image-text matching scores without any training (§3.1), and ii) training the models in the conventional VQA setting without any guidance (§3.2). We obtain a zero-shot overall accuracy of 58.52% which improves to 68.30% when trained without guidance for the CLIP model. In comparison the BLIP-2 model shows a zero-shot accuracy of 64.98% and without guidance accuracy of 75.02%.

The lower part of Table 1 shows the results with various kind of guidances. Most of the guidances improve performance when used individually on their own. For individual guidance, we obtain the highest performance with rationales for both CLIP

| Mode | ScienceQA | | VSR | | IconQA | |
|---|---|---|---|---|---|---|
| | CLIP | BLIP-2 | CLIP | BLIP-2 | CLIP | BLIP-2 |
| No Guidance | 85.32 | 84.28 | 63.99 | 76.76 | 82.36 | 86.21 |
| LG-VQA | | | | | | |
|   Captions | 86.22 | 84.88 | 64.10 | 76.68 | 83.98 | 86.47 |
|   Scene Graph | 86.91 | 86.32 | 63.75 | 77.00 | 82.54 | 86.18 |
|   Objects | **87.22** | 85.28 | 63.95 | 77.40 | 83.25 | 86.52 |
|   CSO | 86.02 | 86.27 | **64.24** | **78.15** | **84.44** | **86.72** |
|   Lecture | 86.92 | 86.27 | - | - | - | - |
|   CSOL | 86.37 | **86.56** | - | - | - | - |

Table 2: Results on the ScienceQA, VSR, and IconQA test sets. CSO denotes captions, scene graphs, and objects guidance. The L in CSOL denotes additional lecture guidance. The lecture guidance is available only for the ScienceQA dataset. Scores are average of three runs.

| Model | Mode | What | Which | Why | How | Where |
|---|---|---|---|---|---|---|
| CLIP | Zero-Shot | 60.14 | 53.33 | 55.74 | 45.00 | 66.0 |
| | No Guidance | 68.96 | 56.00 | 77.05 | 58.33 | **86.0** |
| | Rationale | 75.60 | 64.00 | **90.16** | 63.33 | **86.0** |
| | All | **76.40** | **68.00** | 88.52 | **66.67** | **86.0** |
| BLIP-2 | Zero-Shot | 66.67 | 49.33 | 72.13 | 50.00 | 74.0 |
| | No Guidance | 75.49 | 60.00 | 88.52 | 65.00 | 90.0 |
| | Rationale | 77.32 | 65.33 | 86.89 | 63.33 | 90.0 |
| | All | **80.53** | **66.67** | **90.16** | **70.00** | **94.0** |

Table 3: Results of CLIP and BLIP-2 with different guidance across question types. Scores are average of three runs.

and BLIP-2. In particular, the improvement over no guidance is close to 6% for CLIP, which is also somewhat expected as the rationale generator BLIP-2 model is trained on the A-OKVQA training set itself. The performance with all the guidances is significantly better compared to no guidance for both models, with 7.6% improvement in CLIP and 4.8% improvement in BLIP-2 in the overall set. We also report the scores for the easy and hard subsets in A-OKVQA. We note that the improvement in all guidance setting comes from improvement in both the easy and hard subsets. Notably, BLIP-2 achieves a hard set accuracy of 70% with all guidance compared to 58.57% with no guidance.

**Results for other datasets**  We report results for the other three datasets in Table 2. For ScienceQA without guidance the CLIP model achieves an accuracy of 85.32%, which is a percent higher than the BLIP-2 accuracy of 84.28%. We observe 1-2 % increase in accuracy with the guidances for both models. The objects guidance in CLIP provides the maximum accuracy of 87.22%.

In VSR, the BLIP-2 models heavily outperform the CLIP models. We observe a minor improvement in performance for CLIP with the various guidances. For BLIP-2, the improvement is close to 1.5%, when we use the captions, scene graphs, and objects (CSO) guidance together.

We observe a similar trend in performance for IconQA. The use of various guidances helps in improving the accuracy of both models. However, the margin of increase is not as high in BLIP-2 as it is in CLIP. We observed 2% increase in accuracy for CLIP and 0.5% in increase in accuracy for BLIP-2.

**Analysis**  For A-OKVQA, we analyze the performance of our models across various wh-question types in Table 3. In A-OKVQA, 'What' type questions are the most prevalent type. The incorporation of all guidance helps both CLIP and BLIP-2 to improve on the 'What' question types in addition to all the other question types. For CLIP, the rationale-based guidance outperforms the no guidance across all the question types. Notably, the 'Why' accuracy jumps to 90% from 77%.

**Error Analysis**  We found some common error patterns from our analysis: (i) questions about very small or tiny objects present in the image are erroneously predicted as the visual encoders of CLIP or BLIP-2 are not sensitive enough to detect those objects; (ii) questions about optical characters embedded in the image are often wrongly predicted (which number birthday is being celebrated? for a cake with the number 10 drawn with cream); (iii) questions about abduction (Given the position of the bat and ball the batter most likely did what?) or visual occlusions (What is likely in front of the rug?) are also incorrectly predicted because of insufficient world knowledge.

## 5   Conclusion

In this work, we presented a simple method for visual question answering with language guidance. Our language guidance consists of rationale and explanations constrained upon the image and the question, image captions, scene graphs, and description of the objects present in the image. We propose an effective strategy of fusing the language guidance with the pre-trained multi-modal CLIP and BLIP-2 models. Our language guidance improves the performance of CLIP by 7.6% and BLIP-2 by 4.8% in the challenging A-OKVQA dataset. We also observe consistent improvement in performance with guidance on the ScienceQA, VSR, and IconQA datasets.

## 6 Limitations

As mentioned before, our proposed model underperforms for specific categories of questions where tiny object recognition or optical character recognition on the images is necessary for answering the questions. We also found that the baseline models CLIP and BLIP-2, and our subsequently developed models with guidance cannot adequately model complex world knowledge and commonsense knowledge for answering difficult questions about the image. In the future, we would like to incorporate explicit world and commonsense knowledge as part of the guidance to improve the VQA models.

## 7 Acknowledgments

We thank the anonymous reviewers for their constructive feedback. We also thank Anubhav Jangra for providing feedback on the paper. This project is supported by the AcRF MoE Tier-2 grant (Project no. T2MOE2008, and Grantor reference no. MOE-T2EP20220-0017) titled: "CSK-NLP: Leveraging Commonsense Knowledge for NLP", and the SRG grant id: T1SRIS19149 titled "An Affective Multimodal Dialogue System". This project was partially funded by a grant from the Michigan Automotive Research Center ("ARC"). Any opinions, findings, and conclusions or recommendations expressed in this material are those of the authors and do not necessarily reflect the views of the Michigan Automotive Research Center.

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

# A Experimental Details and Resources

We train our models with the AdamW optimizer (Loshchilov and Hutter, 2017) with a batch size of 8 and 8 epochs. We used a learning rate between {1e-6, 3e-6, 5e-6}. We train our models on a single RTX A6000 GPU. The main task training on A-OKVQA takes around 4 hours for CLIP and 8 hours for BLIP-2 with 8 epochs.