# OpenReview forum: "Language Guided Visual Question Answering: Elevate Your Multimodal Language Model Using Knowledge-Enriched Prompts"
_EMNLP/2023/Conference — EMNLP 2023 Findings_

### Official Review · Reviewer_WdmU · 2023-07-31

**Paper Topic And Main Contributions:** 1. This work contributes a way of inf…
**Soundness:** 2

**Excitement:**

2: Mediocre: This paper makes marginal contributions (vs non-contemporaneous work), so I would rather not see it in the conference.

**Reasons To Accept:**

1. The proposed language guidance improves two popular VQA baselines, CLIP and BLIP-2, for the recent A-OKVQA benchmark.

**Reasons To Reject:**

1. The technical novelty of this work is unclear.
(1) The proposed idea sems to simply supplement the outputs of existing techniques (e.g. RelTR for scene graphs and UniDet for detected objects) into the text input of VQA models, CLIP and BLIP-2.
(2) It is not a surprisingly novel idea in recent popular uses of zero-shot models.

2. Experimental evaluation is highly limited.
(1) This work deals with only two baselines, CLIP and BLIP-2. Admittedly, they are two of the most popularly used vision and language models. However, in order to clearly justify the effectiveness of the proposed guidance, more thorough empirical results with more baselines are required.
(2) This work does not compare with SOTA models for the A-OKVQA benchmark. For example, the proposed guidance can be applied to InstructBLIP, and see whether it improves InstructBLIP too.
(3) Only a single multi-choice question-answering dataset (i.e., A-OKVQA) is tested. It is limited to conclude that the proposed idea works generally.


**Reproducibility:**

3: Could reproduce the results with some difficulty. The settings of parameters are underspecified or subjectively determined; the training/evaluation data are not widely available.

**Reviewer Confidence:**

4: Quite sure. I tried to check the important points carefully. It's unlikely, though conceivable, that I missed something that should affect my ratings.

---

> ### Author Rebuttal · Authors · 2023-08-29
>
> Thanks for your review of the work. Please find the responses below:
>
> **Results on other datasets:**
>
> We experimented on some other datasets -- **ScienceQA (with image context subset), Visual Semantic Reasoning (VSR), and IconQA**. We used the captions, scene graphs, and objects (CSO) as guidance. The ScienceQA dataset also provides a lecture context that can be used as additional guidance. The test set accuracies are as follows:
>
> | Dataset and Guidance |  CLIP | BLIP-2 |
> |----------------------|-------|--------|
> | Science QA           |       |        |
> |    No Guidance       | 85.32 |  84.28 |
> |    Lecture           | 86.02 |  86.12 |
> |    CSO               | 86.02 |  86.27 |
> |    CSO + Lecture     | **86.37** |  **86.56** |
> |----------------------|-------|--------|
> | VSR                  |       |        |
> |    No Guidance       | 63.99 |  76.76 |
> |    CSO               | **64.24** |  **78.15** |
> |----------------------|-------|--------|
> | IconQA               |       |        |
> |    No Guidance       | 82.36 |  86.21 |
> |    CSO               | **84.44** |  **86.72** |
> |----------------------|-------|--------|
>
> The scores are an average of three runs.
>
> **Results of other baselines:**
>
> We performed more experiments with the InstructBLIP Vicuna-7B model and found the results as specified below:
>
> AOKVQA: We were unable to reproduce the original results reported in the InstructBLIP paper with the Vicuna-7B model. However, in our experimental setup, the use of guidance improved the results by 4.6%.
>
> | Setup                    |  InstructBLIP Vicuna 7B |
> |--------------------------|-------------------------|
> | Original Reported Result |         75.7            |
> | Our Reproduced Result    |         72.8            |
> |--------------------------|-------------------------|
> |   + All  Guidance        |         **77.4**        |
> |--------------------------|-------------------------|
>
>
> We also benchmarked InstructBLIP Vicuna-7B on ScienceQA (with image context subset), Visual Semantic Reasoning (VSR), and IconQA datasets. We were not able to exactly reproduce the zero-shot results on these datasets as originally reported in the InstructBLIP. In Science QA we obtained slightly lesser scores than what was originally reported and in VSR, IconQA we obtained higher scores than what was originally reported. Instruct-BLIP used a restricted vocabulary method for calculating the log-likelihood of the answer candidates for the multi-choice question answering tasks. However, some of the key details were unclear to us (restriction over single or multiple tokens, use of temperature, etc.) We thus had our own assumptions which might explain the difference in the reproduced results.
>
> In ScienceQA instances, a lecture context is provided that can be helpful to answer the question. The original results and our reproduced results use this lecture context to obtain accuracies of 60.50 and 59.79, respectively.
>
> In the no guidance setup, we do not use this lecture which results in a drop of accuracy to 54.98. After using the captions, scene-graph, and objects (CSO) guidance the zero-shot accuracy increases to 55.94. Additionally, if we fine-tune only the Q-Former and keep the rest of the model frozen, then the accuracy improvement is observed to be over 5%. **Note that we do not use the Lecture context at all in the CSO guidance setup.** So a steady and consistent performance improvement is observed using our proposed guidance strategies.
>
> | Dataset and Guidance  |  Zero-Shot | QFormer FineTune |
> |-----------------------|------------|------------------|
> | Science QA            |            |                  |
> |  Originally Reported  |    60.50   |       n/a        |
> |  Our Reproduced Result|    59.79   |       62.05        |
> |  No Guidance          |    54.98   |      57.09       |
> |  CSO Guidance         |    **55.94**   |      **62.29**       |
> |-----------------------|------------|------------------|
> | VSR                   |            |                  |
> |  Originally Reported  |    54.30   |       n/a        |
> |  Our Reproduced Result|    59.35   |      60.06       |
> |  CSO Guidance         |    **62.58**   |      **64.52**       |
> |-----------------------|------------|------------------|
> | IconQA                |            |                  |
> |  Originally Reported  |    43.10   |       n/a        |
> |  Our Reproduced Result|    51.12   |      54.68       |
> |  CSO Guidance         |    **52.91**   |      **55.19**       |
> |-----------------------|------------|------------------|
>
> For VSR and IconQA, we observe that the CSO guidance improves over our reproduced results (which is the setup without using any guidance). We observe this improvement in performance in both the zero-shot and QFormer finetune settings.
>
> All the reported scores are an average of three runs.

---

### Official Review · Reviewer_PMF9 · 2023-08-01

**Soundness:** 3

**Excitement:**

3: Ambivalent: It has merits (e.g., it reports state-of-the-art results, the idea is nice), but there are key weaknesses (e.g., it describes incremental work), and it can significantly benefit from another round of revision. However, I won't object to accepting it if my co-reviewers champion it.

**Paper Topic And Main Contributions:**

This paper proposes a multimodal framework that uses language guidance generated by Sota models to answer questions more accurately. The most contribution of this paper is to prove that VQA accuracy can be improved if more information is provided.

**Questions For The Authors:**

1. I want to see if the caption of the image can improve the VQA accuracy, and the compared result with guidance generated by BLIP2.
2. If you ask the model step by step, for example, you first ask the model to generate captions and then ask the model to answer the question, if the accuracy can be improved?
3. Why did you choose CLIP as your model to generate guidance, as far as I know, CLIP is not a model designed for text generation.

**Reasons To Accept:**

This paper uses guidance to improve VQA accuracy.

**Reasons To Reject:**

1. Visualization results should be provided.
2. More baseline models (such as ofa, flamingo) and datasets (such as visual 7w) should be tested.

**Reproducibility:**

4: Could mostly reproduce the results, but there may be some variation because of sample variance or minor variations in their interpretation of the protocol or method.

**Reviewer Confidence:**

3: Pretty sure, but there's a chance I missed something. Although I have a good feel for this area in general, I did not carefully check the paper's details, e.g., the math, experimental design, or novelty.

---

> ### Author Rebuttal · Authors · 2023-08-29
>
> Thanks for your positive review of our work. Please find the responses below:
>
> **Reject 1:** Yes, we will add examples and visualizations in the paper, given the additional page if the paper is accepted. We couldn’t add them to the original submission due to space constraints.
>
> Some examples:
> **Image**: https://cocodataset.org/#explore?id=163640
> **Question**: What type of pants is the man on the right wearing?
> Choices: [linen, corduroy, silk, denim]
> Correct Answer: denim
> **Guidances**:
> Captions: men posing in front of a door holding a snack
> Objects: three person, one bowl, one cabinet/shelf, one hat, one picture or frame, one plate
> Rationales: the man is wearing a pair of blue jeans
>
> **Image**: https://cocodataset.org/#explore?id=182202
> **Question**: What type of device is sitting next to the laptop?
> Choices: [mouse, mobile phone, pen, keyboard]
> Correct Answer: mobile phone
> **Guidances**:
> Captions: a silver laptop on a desk
> Objects: two mobile phone or cell phone, one computer keyboard, one laptop
> Rationales: there is a silver mobile phone on the desk next to the laptop
>
> **Reject 2:** Thanks for the suggestion. We will add the results of more baseline models on the Visual7W dataset in our updated draft. Due to computational constraints and the time limitation of the rebuttal period, we are unable to provide the results now. We promise to include the results on Visual7W in the camera ready.
>
> However, we experimented on some other smaller datasets -- **ScienceQA, Visual Semantic Reasoning (VSR), and IconQA** --- with captions, scene-graph, and objects (CSO) guidance. The ScienceQA dataset also provides a lecture for each question that can be used as a guidance. The test set accuracies are as follows:
>
> | Dataset and Guidance |  CLIP | BLIP-2 |
> |----------------------|-------|--------|
> | Science QA           |       |        |
> |    No Guidance       | 85.32 |  84.28 |
> |    Lecture           | 86.02 |  86.12 |
> |    CSO               | 86.02 |  86.27 |
> |    CSO + Lecture     | **86.37** |  **86.56** |
> |----------------------|-------|--------|
> | VSR                  |       |        |
> |    No Guidance       | 63.99 |  76.76 |
> |    CSO               | **64.24** |  **78.15** |
> |----------------------|-------|--------|
> | IconQA               |       |        |
> |    No Guidance       | 82.36 |  86.21 |
> |    CSO               | **84.44** |  **86.72** |
> |----------------------|-------|--------|
>
>
> **Question 1:** We are unsure what you meant. In the 6th row of Table 1, we have already provided the results when only the caption is used as the guidance. The VQA accuracy is improved when the caption is used for both CLIP and BLIP-2.
>
> **Question 2:** We have only used the (image, text) matching contrastive CLIP and BLIP-2 models as the main VQA model in our paper. We agree with you that the caption generator BLIP-2 model can also be used to generate both the caption and the answer. We think that this setting could be more useful for generative QA tasks, as opposed to the multi-choice QA tasks that we have reported in the paper.
>
> We have tried a very similar setting with the InstructBLIP Vicuna model where the guidance (including captions) are provided in the input to the model. The model sees the (image, question, caption + other guidances) and generates the answer. As Vicuna is a decoder-only model, this is equivalent to your proposed setup of generating the caption and then using the caption to generate the answer.
> We found that the InstructBLIP model is significantly outperformed by the contrastive BLIP-2 models on all the datasets:
>
> | Dataset              |  InstructBLIP Generative | BLIP-2 Contrastive |
> |----------------------|--------------------------|--------------------|
> | AOKVQA               |          77.40           |    **79.83**       |
> | Science QA           |          62.29           |    **86.20**       |
> | VSR                  |          64.52           |    **78.15**       |
> | IconQA               |          55.19           |    **86.72**       |
>
>
>
> **Question 3:** We have not used CLIP for guidance generation. The different guidelines are generated using the various generative models as specified in Sections 3.3 and 3.4.
>
> The CLIP model is only used for the main VQA task (Table 1), where we assume that the different guidelines are provided to it. We show that the CLIP model can effectively use the guidance to improve the VQA accuracy.
>
> All the new reported scores are an average of three runs.

---

### Official Review · Reviewer_ssgE · 2023-08-05

**Soundness:** 4

**Excitement:**

4: Strong: This paper deepens the understanding of some phenomenon or lowers the barriers to an existing research direction.

**Missing References:**

Some relevant related work that might be good. to look at:
- https://openaccess.thecvf.com/content/CVPR2021/papers/Marino_KRISP_Integrating_Implicit_and_Symbolic_Knowledge_for_Open-Domain_Knowledge-Based_VQA_CVPR_2021_paper.pdf
- https://arxiv.org/abs/1909.04696 -EMNLP 2019
- https://arxiv.org/abs/2001.06927 - CVPR 2020

**Paper Topic And Main Contributions:**

The authors use contextual guidance in addition to the image and text to improve performance on VQA that requires outside knowledge to answer. The contextual guidance comes from VLM models tuned on rationales for answering the question in the OK-VQA dataset and from scene graphs predicted from the images. They train the pipeline on the A-OKVQA dataset.

**Questions For The Authors:**

How many parameters/layers are tuned in the BLIP when fine-tuning to generate rationales?
Look at the questions above.

**Reasons To Accept:**

1. Shows evidence in experiments that the extra context helps improve performance for VQA.
2. Nice ablations are done with the type of guidance and the type of questions.

**Reasons To Reject:**

1. The approach requires rationale supervision, which is expensive and might not scale. Authors train and test on the A-OK-VQA dataset, so it is unclear how the rationales trained on that scale on other VQA datasets.
2. Related to the above, it would be interesting to see how this works if you do not train the rationale generation on the OK-VQA dataset. Is there enough knowledge the frozen VLMs to generate a rationale zero-shot that helps for the task?
3. Some qualitative results of the kind of guidance the BLIP-2 would be nice.


**Reproducibility:**

4: Could mostly reproduce the results, but there may be some variation because of sample variance or minor variations in their interpretation of the protocol or method.

**Reviewer Confidence:**

4: Quite sure. I tried to check the important points carefully. It's unlikely, though conceivable, that I missed something that should affect my ratings.

**Typos Grammar Style And Presentation Improvements:**

Minor: line 234 - "text_guide" and "txt_guide" mismatch in equation 7, and the description above it

---

> ### Author Rebuttal · Authors · 2023-08-29
>
> Thanks for your positive review of our work. Please find the responses below:
>
> **Reject 1:** Yes, the rationales generated using the AOK-VQA dataset are key towards the improvement in performance. We have also tried using all the other components except the rationale as the guidance. We achieved the following results:
>
> | Mode                |  CLIP | BLIP-2 |
> |---------------------|-------|--------|
> | No Guidance         | 68.30 |  75.02 |
> | All except Rationale| 71.28 |  78.43 |
> | All                 | **75.98** |  **79.83** |
>
> We observe around 3% improvement over the baseline if we use all components except the rationales for guidance. For other VQA datasets, the explanations as annotated in Li et al., 2018, and Park et al., 2018 (line 195) could be as crucial as the guidance. We will benchmark the VQA v2 dataset and report the results in the updated draft of our work.
>
> **Reject 2:** We tested the frozen InstructBLIP model to generate the rationales in a zero-shot setting and found that it is substantially less suitable for the task. We achieved a ROUGE-L score of 42.9 w.r.t to the gold rationales with our trained rationale generator model in the paper. In comparison, the ROUGE-L score is 24.2 for the InstructBLIP zero-shot setting.
>
> **Reject 3:** We couldn’t add them to the paper due to page limitations. We will add examples and visualizations in the paper, given the additional page if the paper is accepted.
>
> Some examples:
> **Image**: https://cocodataset.org/#explore?id=163640
> **Question**: What type of pants is the man on the right wearing?
> Choices: [linen, corduroy, silk, denim]
> Correct Answer: denim
> **Guidances**:
> Captions: men posing in front of a door holding a snack
> Objects: three person, one bowl, one cabinet/shelf, one hat, one picture or frame, one plate
> Rationales: the man is wearing a pair of blue jeans
>
> **Image**: https://cocodataset.org/#explore?id=182202
> **Question**: What type of device is sitting next to the laptop?
> Choices: [mouse, mobile phone, pen, keyboard]
> Correct Answer: mobile phone
> **Guidances**:
> Captions: a silver laptop on a desk
> Objects: two mobile phone or cell phone, one computer keyboard, one laptop
> Rationales: there is a silver mobile phone on the desk next to the laptop
>
> **Question 1:** The visual encoder and the language model are kept frozen for the rationale generation task. We only train the Q-Former component of the BLIP-2 model (lines 196-201). This results in a total of ~107 million trainable parameters.
>
> **Missing References:** Thanks for the pointers. We will include them in the related work section.

---

### Meta-Review · Area_Chair_C8fc · 2023-09-19

**Recommendation:** 3

**Metareview:**

This paper focuses on knowledge-augmented VQA, which requires common sense knowledge and reasoning among VQA problems. The paper aims to improve the performance of VQAs by providing a variety of relevant knowledge in the form of language guidance; performance gains have been observed when language guidance is introduced for CLIP and BLIP-2-based baselines.

Pros:
Introducing a variety of relevant knowledge in the form of language guidance is a convincing approach.
Performance improvements have been observed.

Cons:
As two reviewers questioned, reviewer PMF9's comments include a misunderstanding, but there are some questions about whether the baseline should only be CLIP and BLIP-2.
As reviewer WdmU points out, there are several baselines and experimental settings, and the range of effects this study demonstrated is limited.
Although Rebuttal shows experimental results for different data sets, it is not a justification for the baseline setting.
CLIP is not good at focusing on the finer points of an image and can only understand the broad features represented in the image. A more detailed discussion is needed regarding what type of instruction is effective. Also, authors may discover cases where the proposal would work better by using a different VQA model (e.g., T5-based) that can capture different image features.

---

### Decision · Program_Chairs · 2023-10-07

**Decision:**

Accept-Findings

**Comment:**

This paper focuses on knowledge-augmented VQA, which requires common sense knowledge and reasoning among VQA problems. The paper aims to improve the performance of VQAs by providing a variety of relevant knowledge in the form of language guidance; performance gains have been observed when language guidance is introduced for CLIP and BLIP-2-based baselines.

Pros:
Introducing a variety of relevant knowledge in the form of language guidance is a convincing approach.
Performance improvements have been observed.

Cons:
As two reviewers questioned, reviewer PMF9's comments include a misunderstanding, but there are some questions about whether the baseline should only be CLIP and BLIP-2.
As reviewer WdmU points out, there are several baselines and experimental settings, and the range of effects this study demonstrated is limited.
Although Rebuttal shows experimental results for different data sets, it is not a justification for the baseline setting.
CLIP is not good at focusing on the finer points of an image and can only understand the broad features represented in the image. A more detailed discussion is needed regarding what type of instruction is effective. Also, authors may discover cases where the proposal would work better by using a different VQA model (e.g., T5-based) that can capture different image features.